# Identification of a DNA Methylation Episignature in the 22q11.2 Deletion Syndrome

**DOI:** 10.3390/ijms22168611

**Published:** 2021-08-10

**Authors:** Kathleen Rooney, Michael A. Levy, Sadegheh Haghshenas, Jennifer Kerkhof, Daniela Rogaia, Maria Giovanna Tedesco, Valentina Imperatore, Amedea Mencarelli, Gabriella Maria Squeo, Eleonora Di Venere, Giuseppe Di Cara, Alberto Verrotti, Giuseppe Merla, Matthew L. Tedder, Barbara R. DuPont, Bekim Sadikovic, Paolo Prontera

**Affiliations:** 1Department of Pathology and Laboratory Medicine, Western University, London, ON N6A 3K7, Canada; Kathleen.Rooney@lhsc.on.ca (K.R.); sadegheh.haghshenas@gmail.com (S.H.); 2Verspeeten Clinical Genome Centre, London Health Sciences Centre, London, ON N6A 5W9, Canada; michael.levy@lhsc.on.ca (M.A.L.); jennifer.kerkhof@lhsc.on.ca (J.K.); 3Medical Genetics Unit, Maternal-Infantile Department, University and Hospital of Perugia, 06129 Perugia, Italy; daniela.rogaia@unipg.it (D.R.); tedesco.mariag@libero.it (M.G.T.); valentina.imperatore@ospedale.perugia.it (V.I.); amedea.mencarelli@unipg.it (A.M.); 4Pediatric Clinic, Department of Medicine, University of Perugia, 06129 Perugia, Italy; giuseppe.dicara@unipg.it (G.D.C.); alberto.verrotti@ospedale.perugia.it (A.V.); 5Laboratory of Regulatory and Functional Genomics, Fondazione IRCCS Casa Sollievo della Sofferenza, 71013 San Giovanni Rotondo, Italy; g.squeo@operapadrepio.it (G.M.S.); e.divenere@operapadrepio.it (E.D.V.); giuseppe.merla@unina.it (G.M.); 6Department of Molecular Medicine and Medical Biotechnology, University of Naples Federico II, 80100 Naples, Italy; 7Greenwood Genetic Center, Greenwood, SC 29646, USA; mtedder@ggc.org (M.L.T.); dupont@ggc.org (B.R.D.)

**Keywords:** 22q11.2 deletion, DNA methylation, episignature, diagnostic method, DiGeorge syndrome, Velocardiofacial syndrome, Conotruncalanomaly face syndrome

## Abstract

The 22q11.2 deletion syndrome (22q11.2DS) is the most common genomic disorder in humans and is the result of a recurrent 1.5 to 2.5 Mb deletion, encompassing approximately 20–40 genes, respectively. The clinical presentation of the typical deletion includes: Velocardiofacial, Di George, Opitz G/BBB and Conotruncalanomaly face syndromes. Atypical deletions (proximal, distal or nested) are rare and characterized mainly by normal phenotype or mild intellectual disability and variable clinical features. The pathogenetic mechanisms underlying this disorder are not completely understood. Because the 22q11.2 region harbours genes coding for transcriptional factors and chromatin remodelers, in this study, we performed analysis of genome-wide DNA methylation of peripheral blood from 49 patients with 22q11.2DS using the Illumina Infinium Methylation EPIC bead chip arrays. This cohort comprises 43 typical, 2 proximal and 4 distal deletions. We demonstrated the evidence of a unique and highly specific episignature in all typical and proximal 22q11.2DS. The sensitivity and specificity of this signature was further confirmed by comparing it to over 1500 patients with other neurodevelopmental disorders with known episignatures. Mapping the 22q11.2DS DNA methylation episignature provides both novel insights into the molecular pathogenesis of this disorder and an effective tool in the molecular diagnosis of 22q11.2DS.

## 1. Introduction

The 22q11.2 deletion syndrome (22q11.2DS) is the most common genomic disorder in humans, with an incidence in about 1/4000 live births [1]. The majority of patients (approximately 90%) have a 2.54-Mb deletion, involving approximately 40 genes, while a further subset of individuals (approximately 7%) have smaller deletions of 1.5 Mb involving about 20 genes and the remaining patients (approximately 3%) have atypical or “nested” deletions [2,3]. 22q11.2DS is associated with multiple clinical presentations, including Velocardiofacial syndrome, Di George syndrome, Opitz G/BBB syndrome and Conotruncalanomaly face syndrome [4]. The major clinical manifestations of 22q11.2DS include: congenital heart disease, particularly conotruncal malformations, palatal abnormalities, immune deficiency, characteristic facial features, hearing loss and learning difficulties. Laryngotracheoesophageal, gastrointestinal, ophthalmologic, central nervous system, skeletal and genitourinary anomalies may also occur. Moreover, psychiatric illness, in particular schizophrenia, Parkinson disease and autoimmune disorders are more common in individuals with 22q11.2DS [3,5,6,7,8]. Due to the broad range of clinical presentations, including those within the same family, molecular testing is required to establish the diagnosis.

The correlation between genotype and phenotype is poorly understood. A connection between CD4+ lymphopenia and deletion breakpoints has been reported [9], but other studies failed to identify a relationship between phenotype and deletion extension. To date, no single gene has been identified to explain all the features of 22q11.2DS, and it is therefore considered a contiguous-gene deletion syndrome. Regarding its complex pathogenesis, some authors [10] have suggested that it is primarily a transcription deregulation syndrome, due to the involvement of different transcription factor genes, such as *TBX1*, and chromatin remodeling genes, such as *HIRA*. Few studies have been performed to explore if an altered methylation profile is present in patients with 22q11.2DS [8] and, to our knowledge, no studies to date have established evidence for a specific genome-wide DNA methylation profile.

Notably, the clinical characteristics of the 22q11.2DS overlap various conditions, including CHARGE, Cornelia de Lange, Koolen-De Vries, Alagille and Kabuki syndromes, all of which are caused by mutations in chromatin remodelling genes and are considered in the differential diagnosis. One functional consequence of genetic defects in patients with hereditary neurodevelopmental disorders can be the disruption of genomic DNA methylation [11]. DNA methylation is an epigenetic modification, resulting in changes in structural and chemical properties of the DNA, impacting molecular mechanisms including chromatin assembly and gene transcription. Our group and others have demonstrated that a growing number of Mendelian genetic disorders exhibit DNA methylation episignatures as highly sensitive and specific biomarkers in the peripheral blood of affected individuals [12,13,14,15,16,17,18,19,20,21]. These genome-wide DNA methylation profiles currently include over 40 rare disorders and represent effective biomarkers for the diagnosis of patients across a broadening range of neurodevelopmental genetic conditions, as well as a reflex functional assay for patients with ambiguous genetic findings or clinical phenotypes [22]. Based on this technology, a genome-wide DNA methylation analysis assay, known as EpiSign, has recently been adapted for the diagnosis of Mendelian disorders [23]. 

In this study, our primary goal was to investigate whether a specific genome-wide episignature exists in patients with the typical 22q11.2 deletion and if this episignature could be used to confirm a clinical diagnosis. Our secondary objective was to evaluate if methylation alteration may impact genes potentially relevant in the pathogenesis of the complex 22q11.2DS phenotype.

## 2. Results

We compiled peripheral blood DNA methylation data generated using Illumina Infinium Methylation arrays for 49 patients with a confirmed clinical and molecular diagnosis of 22q11.2DS. The clinical features of each patient and the coordinates of deletion breakpoints are depicted in Appendix A. The cohort consisted of 29 males and 20 females, with an age range of approximately 1 month to 41 years. 

The deletions ranged in size from 406 kb to 3 Mb, and 43 cases are definable as typical deletions. Cases 12 and 30 are proximal deletions and cases 15, 21, 41 and 45 are distal deletions (Figure 1).

Following quality control and probe selection, using the training cohort of 49 patients, 160 differentially methylated CpG probes were retained for 22q11.2DS episignature discovery (Appendix A) and represent the most differentially methylated CpGs in the training cohort when compared to controls Hierarchical clustering (heatmap) demonstrated that the selected probes were able to completely separate the 22q11.2DS cases from age and sex matched controls (Figure 2A) as did MDS (Figure 2B), confirming the robustness of the episignature. In addition, these cases remain segregated from the control samples contained in both the same batch and different batches (non-batch), confirming that batch structure is not contributing to the episignature (Figure 2). We also assessed sample distribution by both age and sex by Euclidean and MDS clustering to determine if sub-clustering occurred (Appendix A). No data sub-structure for age or sex was observed. 

Leave-1-out and leave-25%-out cross validation, using MDS and hierarchical clustering, provided further evidence of the robustness and sensitivity of the signature. In all rounds, the testing case(s) was correctly clustered with the training case samples. In addition, the SVM classification model was applied to each round of testing to predict the accuracy of the MVP classifier (Appendix A). A summary of the MVP scores from leave-1-out cross validation results from each round of classification can be seen in Figure 3A. MVP scores show that, in most instances, the MVP score obtained was ≥0.75. This confirms the sensitivity of the signature to identify 22q11.2DS cases.

Next, we constructed a binary SVM classifier using the selected probes from the training cohort (22q11.2DS_training) and matched controls. The MVP score for each case was near 1, indicating similarity of the observed methylation pattern to the 22q11.2DS episignature (Figure 3B). All cases were classified correctly with no overlap with syndromes in the differential diagnosis, including Koolen-De Vries (KDVS), Kabuki, Cornelia de Lange (CdLS) and CHARGE. This analysis demonstrates high-level of specificity for the 22q11.2DS episignature. 

Next, we assessed the segregation of the validation cohort and the 4 distal 22q11.2DS cases against the episignature. We performed hierarchical clustering, MDS (Figure 4), and applied the SVM model (Figure 4C). Collectively, these show that distal cases 15, 21, 41 and 45 do not exhibit a similar DNA methylation signature to the training cohort. This also confirms that all 22q11.2DS validation cases segregate with the episignature, further confirming the sensitivity of the classifier. 

### 2.1. Screening Unresolved Cases

Using the SVM classifier in Figure 4C, all unresolved cases in the EKD were screened against the new 22q11.2DS episignature. Two cases showed high MVP scores (both close to 0.75) and segregated with or close to cases in subsequent hierarchical clustering and MDS plots (Figure 5). Through follow up with the submitting centres, we were able to confirm one case (Figure 5A), a male aged 2 years, as carrying a deletion of 22q11.2q11.21 (17041749_19835388) with a clinical diagnosis of 22q11.2 microdeletion syndrome. The second case (Figure 5B), a male aged 5 years, carried the variant UBE2A:NM_003336: c.283C > T:p.Arg95Cys with a clinical diagnosis consistent with X-linked Mental retardation, Nascimento-type (MRXSN) (OMIM#300860). Previous analysis showed that this case did not segregate with the MRXSN episignature. It is possible that this individual may have an underlying copy number variant in the 22q11.2 region; however, microarray follow up remains outstanding at this time. 

### 2.2. Differentially Methylated Regions

Using the DMRcate algorithm [24] with 5% or 10% mean methylation difference and 3 or 5 CpGs, we identified regions of differential methylation (Table 1). Three and 5 CpGs were selected as a less or more stringent minimum analysis, respectively, with all CpGs falling within a maximum of 1000 nucleotides in order to be considered a DMR [24]. These regions are located across multiple chromosomes, exhibiting significant overlap with genes or regulatory elements. A full list of DMRs can be found in Appendix A. 

## 3. Discussion

The 22q11.2DS is the most frequent genomic disorder and among the most clinically variable condition in humans. It can often be strongly suspected on clinical findings, particularly when facial gestalt is present and the clinician has expertise in dysmorphology, or when many typical signs and symptoms co-exist in the same patient (for example, conotruncal heart malformation and palate defect). However, in many cases the diagnosis requires genetic testing. Many patients are diagnosed using chromosomal microarray (CMA) as part of the diagnostic screening in patients with non-specific phenotypes, including hypotonia, developmental disorders (DD), intellectual disability (ID) and/or multiple congenital anomalies (MCA) [1,2,3,4]. The process leading to an aetiologic diagnosis in patients with ID/DD/MCA is frequently complex and includes the interaction between expert clinicians and biological information gained from metabolic, chromosomal, genomic, sequencing and methylation profile tests. Deciding on the most appropriate genetic test for a specific patient can be difficult when the clinical phenotype is nonspecific and many other disorders have overlapping features that are included in the differential diagnosis, such as chromatinopathies.

Genome-wide DNA methylation analysis has been used to identify specific episignatures in more than 40 genetic diseases, including some common microdeletion/duplication syndromes [12,13,14,15,16,17,18,19,20,21]. In addition to syndromes involving genomic imprinting centres (PWS/AS, Beckwith–Wiedemann, Silver–Russell), methylation arrays have been shown to detect other microdeletion syndromes including 7q11.23 deletion syndrome (Williams–Beuren syndrome) [25] and 22q13.3 deletion syndrome (Phelan McDermid syndrome) [26].

In this study, we applied a previously developed methodology [12,16,26,27] to analyse 49 patients with 22q11.2DS (Appendix A). The methylation profile of 45 22q11.2DS patients, including all 43 typical 2.54 Mb and 1.5 Mb deletions, and the 2 proximal deletions, differs significantly from normal controls (Figure 4) and other neurodevelopmental disorders (Figure 4C). The 4 patients showing methylation profiles that differ from the 22q11.2DS episignature carry atypical, distal deletions (patients 15, 21, 41, 45, Appendix A), suggesting the presence of two distinct clinical entities, at least on the basis of an episignature. In fact, these findings coincide with the evidence of a more similar phenotype among patients carrying the typical 2.54 Mb deletion with those harbouring the 1.5 Mb deletion, than to patients with distal deletions that usually have a milder or no phenotype [28]. Notably, three (patients 15, 21, 45) out of four of the distal deletion cases are maternally inherited, while the fourth case (patient 45) also carries a de novo 13q12.2q12.3 deletion that may be responsible for his phenotype. In contrast, all of the typical 22q11.2 deletion cases are de novo events (Appendix A).

The differentially methylated regions (DMRs) are not located within the chromosome 22, and therefore the haploinsufficiency of genes in the 22q11.2 region alters the methylation profile at the genome level, consistent with other chromathinopathies. Using the DMRcate algorithm [24], we identified 5 regions of differential methylation with 10% difference across 3 CpGs, 329 regions with 5% difference across 5 CpGs and 1063 regions with 5% difference involving 3 CpGs (Table 1 and Appendix A). These regions are located across multiple chromosomes and may involve clinically relevant genes or regulatory elements. 

The NFIA gene is one of the few genes with a 10% methylation difference, and the hypomethylated CpG probes are located in the promoter region of the gene (Appendix A). Loss of function variants in NFIA are responsible for brain malformation with or without urinary tract defects (OMIM#613735), and patients with *NFIA* variants show dysmorphisms like small chin, dysplastic helices, small mouth, and other features such as hypoplastic kidney, hydronephrosis, hypotonia, global developmental delay, intellectual disability or agenesis of corpus callosum, similar to patients with 22q11.2DS. While we observed hypomethylation at the NFIA promoter, which is often associated with increased gene expression, the function of NFIA in neurodevelopment is temporal as it is required for gliogenic switch and is also crucial for the maintenance of the undifferentiated progenitor pool [29,30]. Additionally, overexpression of NFIA prevents further differentiation into astrocytes until the gene is downregulated [30]. Therefore, altered or inappropriate expression during development may contribute to the intellectual disability phenotype of 22q11.2DS patients, but further mechanistic and expression studies are required. Many other genes (Appendix A) appear to be potentially deregulated by methylation defects, specifically hypomethylation, that may contribute to the phenotype of 22q11.2DS, such as: *HOXA2* (promoter region), whose variants are associated with microtia, hearing impairment and cleft palate (OMIM#612290); *IRF8* (gene body), associated with immunodeficiency 32B (OMIM#226990); *ANKR11D* (gene body), responsible for KBG syndrome characterized by hypoplastic alae nasi, short stature, skeletal abnormalities and intellectual disability (OMIM:#148050). While these findings demonstrate evidence of reproducible methylation changes in or near gene promoters, additional studies, including gene expression profiling, will be necessary to determine the functional impact of these changes on the downstream molecular mechanisms and the related pathophysiology. A larger cohort of patients and expression studies will be useful to clarify these aspects; however, what emerged from these results is that the phenotype of 22q11.2DS could be the summation of haploinsufficiency of 22q11.2 genes as well as global and specific methylation defects. 

The evidence of such a specific episignature underlies the presence of a common process in the alteration of chromatin remodelling. Looking at the minimal region of overlap among all 45 deletions sharing this specific episignature, we can note that patient 30 (Figure 1) is the more proximal deletion identified that does not overlap with the others deletions. However, this patient is the son of a mother carrier of a balanced translocation between chromosome 13 and 22, and he shows the karyotype 46,XY,der(13)t(13;22)(q21.3;q11.2)+13[mat], which arose from the unbalanced translocation. The subsequent array disclosed the small proximal deletion on 22q11.2 but also the partial trisomy of chromosome 13, approximately 53 Mb in size. Therefore, we have at least two possible explanations for the episignature of this patient: the first is the possible disruption of enhancer or regulatory regions distal to the deletion and thus involving other 22q11.2 genes, the second is the possible presence of a more wide and complex methylation alteration related to partial trisomy 13 that overlap those of 22q11.2DS and that we, at this time, are not able to identify because we do not have a specific cohort of patients with whole chromosome or partial trisomy 13. Additional methylation and transcriptional studies in patients with trisomy 13 would be required to further elucidate this finding. In fact, considering the other 44 deletions with a common episignature, we found the following genes in the minimal region of overlap, spanning about 397 Kb (chr22:19004735-19401630): *DGCR9, DGCR10, DGCR11, DGCR2, DGCR14, TSSK2, GSC2, SLC25A1, CLTCL1, HIRA*. Excluding from this analysis the more proximal deletion (patient 12) and considering only the typical deletions, the number of genes increases and also includes: *C22orf39, CDC45, CLDN5, SEPT5, GP1BB TBX1, GNB1L, C22orf29, COMT, MIR4761, TXNRD2, ARVCF, TANGO2, TRMT2A, DGCR8, MIR1306, RANBP1.* Among these genes, we identified the following as interesting for their biological functions, in relation to transcription and methylation activities: *HIRA, TBX1, COMT* and *TRMT2A*.

*HIRA* (OMIM#600237) encodes a histone chaperone that deposits the histone variant H3.3 in transcriptionally active genes. It regulates neural progenitor proliferation and neurogenesis, and it belongs to the WD40 Repeat (WDR) protein family involved in brain development and neuronal connectivity. Recent studies on *HIRA* knock-down mouse models demonstrated that its haploinsufficiency is associated with abnormal neurodevelopment and impaired dendritic outgrowth [31]. Other authors analysed hematopoietic cells with specific *HIRA* deletion in mice and showed that this dramatically reduces bone marrow hematopoietic stem cells (HSCs), resulting in anaemia, thrombocytopenia and lymphocytopenia [32]. The T-box transcription factor *TBX1* (OMIM#602054) is known to be involved in the regulation of developmental processes, including those responsible for development of the pharyngeal arches [33], and this has led to the hypothesis that it may be responsible for some of the features typically seen in 22q11.2DS, including cleft palate and conotruncal anomaly face [34]. Interestingly, there are two methyltransferases contained within the common region of overlap of the typical cases: the catechol-o-methyltransferase *COMT* (OMIM#116790) and the tRNA methyltransferase 2 homolog *TRMT2A* (OMIM#611151). The *COMT* gene plays a role in dopamine metabolism and has been postulated as a strong candidate for susceptibility to schizophrenia, a feature observed in patients with Velocardiofacial syndrome [35,36]. Methyltransferases have been shown to be responsible for episignatures in other neurodevelopmental disorders, e.g., *DNMT3A* in patients with Tatton–Brown–Rahman syndrome [12] and *KMT2D* in Kabuki syndrome [37].

However, to fully delineate the involvement of specific genes that could be responsible for the episignature described here in 22q11.2DS patients, it would be prudent to assess the methylation profiles of patients with point mutations in genes such as *TBX1*, *COMT, TRMT2A* and *HIRA*. In the future we are planning to analyse, genome-wide DNA methylation profiles in patients with atypical deletions, as this may expand or refine the episignature and may help in determining the genes related to epiphenomena. 

## 4. Material and Methods

### 4.1. Study Cohort

The patients were admitted to the genetic counselling services at Medical Genetics Unit, Hospital and University of Perugia, Italy, and Greenwood Genetics Centre, USA, for genetic consultation as part of routine clinical assessment. The study was approved by the Western University Research Ethics Board (REB 106302, 10 August 2020). Written informed consent for publication of clinical data was obtained from the patients’ families. We enrolled in this study, 49 individuals for which a diagnosis of 22q11.2DS had been previously made; by array-CGH, and in some cases confirmed by a second method (FISH, MLPA, second array-CGH), as previously described [38,39]. Forty-three patients, with typical 2.54 Mb or 1.5 Mb deletions, showed clinical presentation compatible with the genetic diagnosis, while the other 6 patients with atypical deletions (4 distal and 2 proximal), presented with an unspecific phenotype, mainly characterized by developmental or language delay (Appendix A). Thirteen typical cases were randomly selected and removed to use as a validation cohort (22q11.2 validation), the remaining 30 typical and 2 proximal cases were used as the training cohort (22q11.2DS_training), and the 4 distal cases were subsequently plotted alongside the mapped signature to determine their similarity (22q11.2DS_distal).

### 4.2. DNA Methylation Experiment

Analysis was carried out on peripheral blood-extracted genomic DNA using standard techniques at London Health Sciences Centre, Canada. Following bisulphite conversion, DNA methylation analysis was performed using the Illumina Infinium Methylation EPIC bead chip arrays (San Diego, CA, USA) according to manufacturer’s protocols. Analysis and episignature discovery were carried out based on our laboratory’s previously published protocols [12,16,26,27]. Intensity data files containing methylated and unmethylated signal intensities were analysed in R 4.0.2. The methylation data was normalised using the Illumina normalization method with background correction using the minfi package [40]. The following probes were eliminated; probes with detection *p* value > 0.01, probes located on chromosomes X and Y, probes containing single nucleotide polymorphisms (SNPs) at or near the CpG interrogation or single nucleotide extension sites and probes that cross react with genomic regions other than their target regions. In addition, samples containing failed probes of more than 5% (*p*-value > 0.1, calculated by the minfi package) were removed. All samples were examined for genome-wide methylation density, and those deviating from a bimodal distribution were excluded. Principal component analysis (PCA) was performed to examine batch structure and identify outliers. Matched controls were randomly selected from the EpiSign Knowledge Database (EKD) [12], matched by age and sex using the MatchIt package. Controls included samples from the same batch as the 22q11.2DS cases to account for batch structure and possible batch effect, and included samples from unaffected controls and other known episignature cohorts within the EKD. 

Methylation levels for each probe (beta values) were converted to M-values by logit transformation using the equation log2(beta/(1 − beta)). A linear regression model was applied to identify differentially methylated probes using the limma package [41]. Estimated blood cell proportions were incorporated into the model matrix as confounding variables [42]. Using the eBayes function in the limma package, the generated *p* values were moderated.

### 4.3. Probe Selection, Clustering and Dimension Reduction

Probe selection was performed in three steps. Firstly, 900 probes were selected with the highest product of methylation differences between cases and control samples and the negative value of the log-transformed *p* value (−log(*p* value)). Secondly, a receiver’s operating characteristic (ROC) curve analysis was performed, and 450 probes were retained with the highest area under the ROC curve (AUC). Lastly, probes with pair-wise correlation greater than 0.60 measured using Pearson’s correlation coefficients for all probes were eliminated. Hierarchical clustering was performed with the remaining probes using Ward’s method on Euclidean distance by the R gplots package. Multidimensional scaling (MDS) was carried out by scaling of the pair-wise Euclidean distances between samples.

### 4.4. Cross Validation and Batch Structure

Leave-1-out cross validation was performed to validate the presence of an episignature. Multiple rounds of analysis were performed, leaving out one case for testing during probe selection. This was repeated until all samples had been tested (left out). Additionally, we performed cross validation, leaving out 25% of samples in each round. All samples were plotted using hierarchical clustering and multidimensional scaling to determine if cases used for training remained segregated with cases that were not (testing samples), and to ensure that all cases remained separate from controls. In addition, we evaluated the methylation patterns of the other samples from the same experimental batch to account for possible batch effect.

### 4.5. Classifier Model and Differentially Methylated Regions

A binary support vector machine (SVM) classifier model with linear kernel was constructed using the e1071 package, as described previously, and used to generate Methylation Variant Pathogenicity (MVP) scores [12,16,18]. This classifier gives a score between 0 and 1 for each sample, which represents the confidence in predicting whether the subject has a DNA methylation profile similar to 22q11.2DS. Conversion of SVM decision values to these scores was done according to the Platt’s scaling method [43]. The training cohort, 75% of unaffected controls and 75% of other samples from 38 additional Mendelian neurodevelopmental disorders was used to train the classifier model. These disorders were included in the classifier to assess the specificity of our classification model for 22q11.2DS compared to other EpiSign disorders [12]. The remaining 25% of unaffected controls and other samples were used as the testing set. To determine whether this model was sensitive to other conditions commonly assessed as a differential diagnosis for 22q11.2DS, we tested over 1500 subjects with confirmed clinical and molecular diagnosis of such neurodevelopmental syndromes with this classification model. These syndromes included Koolen-De Vries, Kabuki, Cornelia de Lange and CHARGE. This classification model was also applied to the leave-1-out and leave-25%-out cross validation, training against only controls, to estimate how accurately the predicted MVP score would perform in testing and determine the sensitivity of the episignature.

Differentially methylated regions (DMRs) were detected using the DMRcate package [24], and regions containing at least 3 CpGs within 1 kb with a minimum methylation difference of 10% and a Fisher’s multiple comparison *p*-value < 0.01 were selected. Detection was repeated for both 5 CpGs and 5% methylation difference.

## 5. Conclusions

In conclusion, our data adds 22q11.2DS to the list of genomic disorders diagnostically detectable by methylation assay and will permit the inclusion of this episignature in the methylation genetic screening tests often carried out in patients with ID/MCA. This episignature is highly specific and can be used to differentiate 22q11.2DS from other syndromes commonly considered in the differential diagnosis due to overlapping clinical manifestations. Moreover, this data contributes additional knowledge in the pathogenesis of 22q11.2DS. Further studies will aim to identify the gene(s) in the region of which haploinsufficiency leads to DNA methylation alteration and how these, in turn, may modify the transcriptional pathway. Expanding the size of patient cohorts with and without specific clinical features, and with differential genomic breakpoints, would be needed to assess evidence of distinct episignatures that may be useful as predictive biomarkers in association with the variable penetrance and expressivity in 22q11.2DS.

## Figures and Tables

**Figure 1 ijms-22-08611-f001:**
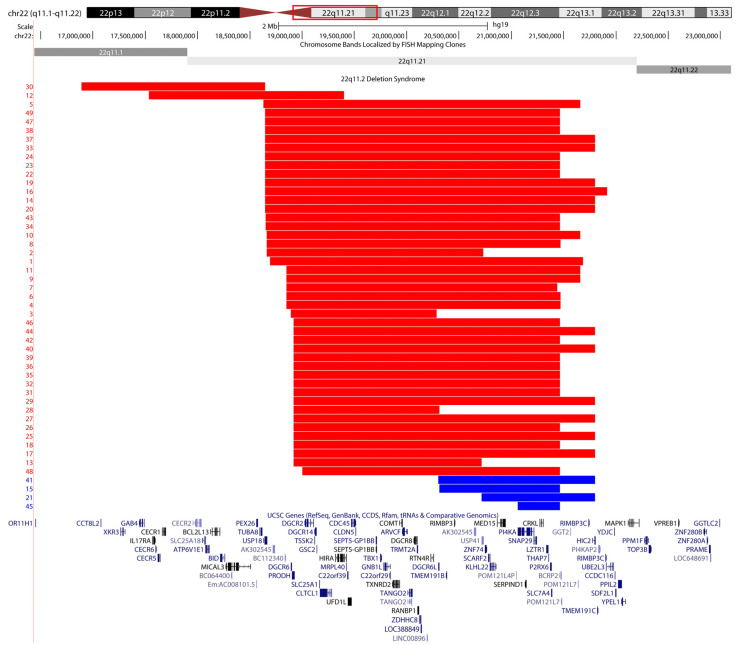
Genomic region of the 22q11.2DS deletions; those with methylation episignature are depicted in red, and the cases showing no episignature in blue. Case 30 does not overlap in coordinates with other cases that exhibit an episignature; however, this patient carries an additional 53 Mb duplication of chromosome 13 and therefore cannot be considered as simply 22q11.2DS. Cytogenetic bands and known genes are presented in this figure using the UCSC genome browser (http://genome.ucsc.edu accessed on 5 August 2021) 2009 (GRCh37/hg19) genome build (Kent 2002).

**Figure 2 ijms-22-08611-f002:**
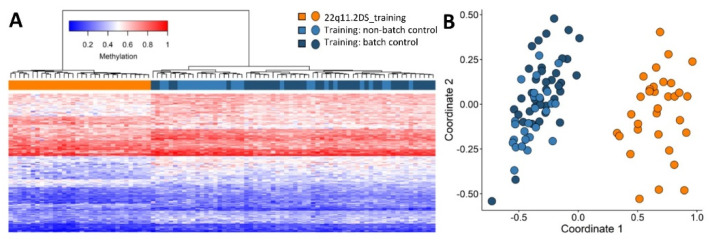
Analysis of robustness of the episignature to differentiate between 22q11.2DS cases from age and sex matched controls. (**A**) Euclidean hierarchical clustering (heatmap); each column represents a single case or control, each row represents one of the 160 CpG probes selected for the episignature. This heatmap shows clear separation between 22q11.2DS cases (orange) from controls, both within-batch (dark blue) and non-batch (light blue). (**B**) MDS plot shows segregation of cases from controls and confirms robustness of the episignature.

**Figure 3 ijms-22-08611-f003:**
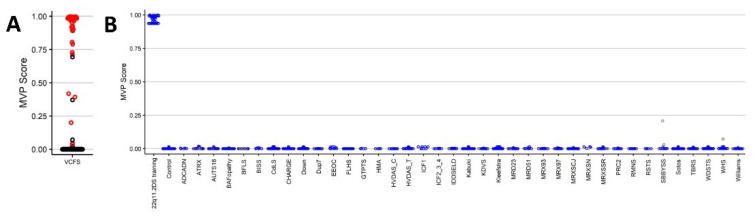
(**A**) Leave-1-out cross validation. Each sample was used for testing once and plotted with the remaining 31 samples used for training. Figure depicts a summary of the MVP classifier used during cross validation. Each 22q11.2DS testing sample is marked in red; all other samples in the classifier are depicted in black. (**B**) SVM classifier model. Model trained using the selected probes from the 22q11.2DS training cases, 75% of controls and 75% of other neurodevelopmental disorder samples (depicted in blue). The remaining 25% of controls and 25% of other disorder samples were used for testing (displayed in grey). Plot shows the 22q11.2DS cases with an MVP score close to 1 compared with all other samples, showing the specificity of the classifier to detect 22q11.2DS samples.

**Figure 4 ijms-22-08611-f004:**
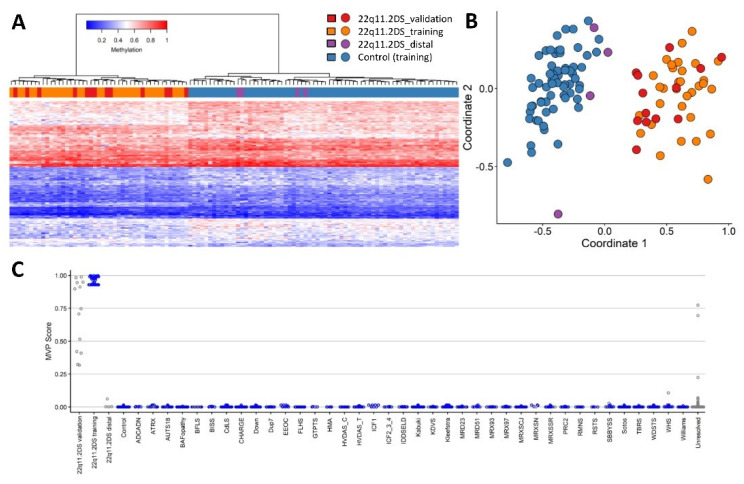
Results of plotting the validation cohort (22q11.2DS_validation) and the 4 distal cases (22q11.2DS_distal) against the episignature. (**A**) Hierarchical clustering shows the validation cases (in red) segregating with training cases (in orange). Heatmap shows the 4 distal deletions (in purple), clearly segregating with controls (blue) and away from the remaining 22q11.2DS cases. (**B**) MDS confirming distal cases segregate closer to controls and validation cases with training cases. (**C**) SVM classifier showing classification of validation and distal cases. No validation cases have an MVP score of zero, and the majority are over 0.50. In contrast, all 4 distal cases have an MVP score at or close to zero.

**Figure 5 ijms-22-08611-f005:**
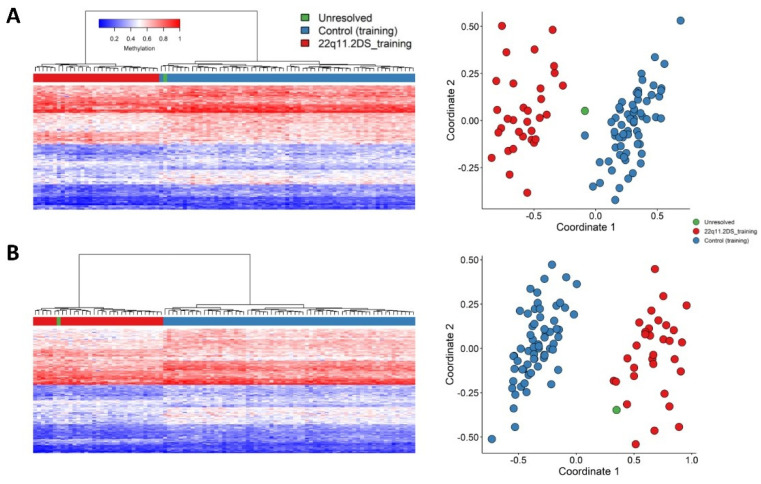
Results of hierarchical clustering and MDS plots for the two unresolved cases in the EKD with high MVP scores for 22q11.2DS episignature, from screening. (**A**) Individual with confirmed molecular and clinical diagnosis of 22q11.2DS (depicted in green). In both the heatmap and MDS, the case lies between 22q11.2DS cases (in red) and controls (in blue). (**B**) Individual with variant in UBE2A and confirmed clinical and molecular diagnosis of MRXSN. Case clearly segregates with 22q11.2DS episignature in the heatmap and MDS.

**Table 1 ijms-22-08611-t001:** Number of differentially methylated regions (DMRs) identified between 22q11.2DS cases and controls.

Number of CpGs	Methylation Difference (%)	Number of DMRs
5	10	0
5	5	329
3	10	5
3	5	1064

## Data Availability

The authors confirm that the data supporting the findings of this study are available within the article and its Appendix A. The raw data that support the findings of this study are available on request from the corresponding authors.

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
