# Peer review of "Identification of a DNA Methylation Episignature in the 22q11.2 Deletion Syndrome"

_ijms, 2021, doi:10.3390/ijms22168611_

Round 1
Reviewer 1 Report
In this work, the authors aimed to identify episignatures in patients with typical 22q11.2 deletion. The analysis was performed on genomic DNA extracted from peripheral blood and using Illumina's EPIC bead chip arrays.
The authors identified many differentially methylated regions associated with 22q11.2 syndrome.
Overall, the paper is well and clearly written. The major weakness of the paper is the lack of experimental validation of the identified methylation signatures.
First, an alternative method such as pyrosequencing should be used to validate 10-20 candidate regions to exclude the false positive calls from the Illumina chip.
Second, in the discussion section, the authors describe candidate genes that could presumably be regulated by identified epimutations. However, there is no information about the CpG location or DNA methylation differences in the genes. Moreover, at least the effect of CpG methylation on mRNA expression of the target genes should be studied.
Minor points:
Line 193. unclear is the number of 160 probes. It is mentioned that samples from 49 patients were analysed.
Line 272. 3 or 5 CpGs were differentially methylated. Unclear. Why not 4 or 2? Please explain the analysis.
Author Response
Author’s comment: We thank the reviewers for their comments and suggestions. Please note that unless otherwise indicated line numbers throughout our responses refer to lines in the revised manuscript with Word Track Changes set to show All Markup. Figure and table numbers also refer to those in the revised manuscript.
Reviewer 1:
Comments and Suggestions for Authors:
In this work, the authors aimed to identify episignatures in patients with typical 22q11.2 deletion. The analysis was performed on genomic DNA extracted from peripheral blood and using Illumina's EPIC bead chip arrays.
The authors identified many differentially methylated regions associated with 22q11.2 syndrome.
Overall, the paper is well and clearly written. The major weakness of the paper is the lack of experimental validation of the identified methylation signatures.
First, an alternative method such as pyrosequencing should be used to validate 10-20 candidate regions to exclude the false positive calls from the Illumina chip.
Author’s response: Thank you for this comment and your suggestion. Our group and others have previously demonstrated a high level of concordance between differentially methylated CpGs detected by whole genome methylation arrays, by pyrosequencing or bisulfite sequencing (ex. PMID: 27602171, “Identification of a methylation profile for DNMT1-associated autosomal dominant cerebellar ataxia, deafness, and narcolepsy”, Clinical Epigenetics, 2016, PMID: 27934915, “The defining DNA methylation signature of Floating-Harbor Syndrome”, Scientific Reports, 2016, PMID: 31639040, “DNA methylation signature is prognostic of choroid plexus tumor aggressiveness”, Clinical Epigenetics, 2019). In this study, as in the majority of the more recent literature in this field (PMID: 34087165, “Evaluation of DNA Methylation Episignatures for Diagnosis and Phenotype Correlations in 42 Mendelian Neurodevelopmental Disorders”, American Journal of Human Genetics, 2020, PMID: 30929737, “Diagnostic Utility of Genome-wide DNA Methylation Testing in Genetically Unsolved Individuals with Suspected Hereditary Conditions”, American Journal of Human Genetics, 2019, PMID: 30459321, “BAFopathies' DNA methylation epi-signatures demonstrate diagnostic utility and functional continuum of Coffin-Siris and Nicolaides-Baraitser syndromes”, Nature Communications, 2018, PMID: 33909990, “Truncating SRCAP variants outside the Floating-Harbor syndrome locus cause a distinct neurodevelopmental disorder with a specific DNA methylation signature”, American Journal of Human Genetics, 2021), high level of reproducibility, sensitivity and specificity of the methylation findings has been demonstrated by assessing independent validation cohort not used to discover/train the episignature, and well as through use of the leave-one-out cross-validation experiments. These methodologies proved a robust, statistically-based method to demonstrate the validity of the methylation findings. Another advantage of this approach is that the methylation levels (characterized by microarray signal beta values), represent a mean of thousands of the repeated measures for each individual CpG locus (individual Infinium methylation probe sets are printed 100’s-1000’s of times per microarray chip). This level of sampling provides the accuracy much beyond the both classical (bisulfite clonal sequencing) and current (bisulfite NGS sequencing) techniques enabling reproducible assessment of methylation changes as low as 5%.
Second, in the discussion section, the authors describe candidate genes that could presumably be regulated by identified epimutations. However, there is no information about the CpG location or DNA methylation differences in the genes. Moreover, at least the effect of CpG methylation on mRNA expression of the target genes should be studied.
Author’s response: As suggested, we have now added additional information in Discussion (lines 329 to 355) to indicate where each DMR is located within the gene, as well as the type of methylation change observed (hypo- or hypermethylation). While the objective of this paper was to demonstrate evidence of a sensitive and specific DNA methylation episignature, although beyond the scope of the current study, we agree that it would be interesting to study correlation of these findings to gene expression and other downstream molecular mechanisms. We have also added following to the discussion section, paragraph 5:
“While these findings demonstrate evidence of reproducible methylation changes in or near gene promoters, additional studies, including gene expression profiling, will be necessary to determine the functional impact of these changes on the downstream molecular mechanisms and the related pathophysiology.”
Minor points:
Line 193. unclear is the number of 160 probes. It is mentioned that samples from 49 patients were analysed.
Author’s response: Thank you for pointing the need to clarify this point. We have added language (results, paragraph 3 and figure 2 legend) to clarify that training cohort contained 49 patients and that the 160 probes retained were the most differentially differentiated probes in that group when compared to controls.
Line 272. 3 or 5 CpGs were differentially methylated. Unclear. Why not 4 or 2? Please explain the analysis.
Author’s response: We used DMRcate method (PMID: 25972926) to identify differentially methylated regions. DMRcate algorithm is commonly used in genomic methylation analysis and is optimized for methylation array data. This algorithm is designed to measure methylation changes involving at the minimum 2 adjacent CpGs, within 1kb. We used the 3 and 5 CpG minimum increments as a more permissive and more stringent cut-off to identify a list of regions with more moderate, and more robust changes respectively. We have clarified this in the text as well (results, paragraph 8).
Reviewer 2 Report
In the classifier model, the authors used large number of subjects with confirmed clinical and molecular diagnosis of neurodevelopmental syndromes. Can you authors elaborate the number of subjects used as comparators for their model?
The age range of the patients selected in this study is quite large, from 1 month to 43 years. This reviewer is curious if the authors observed any evidence of differential degree of expression/occurrence of this epigenetic signature with age or sex?
Author Response
Reviewer 2:
Comments and Suggestions for Authors
In the classifier model, the authors used large number of subjects with confirmed clinical and molecular diagnosis of neurodevelopmental syndromes. Can you authors elaborate the number of subjects used as comparators for their model?
Author’s response: Thank you for this comment and suggestion. We have provided further details in the methods “Classifier Model and Differentially Methylated Regions” section:
“To determine whether this model was sensitive to other conditions commonly assessed as a differential diagnosis for 22q11.2DS, we tested over 1500 subjects which are part of the EpiSign Knowledge Database (EKD) with confirmed clinical and molecular diagnosis of such neurodevelopmental syndromes with this classification model”
The age range of the patients selected in this study is quite large, from 1 month to 43 years. This reviewer is curious if the authors observed any evidence of differential degree of expression/occurrence of this epigenetic signature with age or sex?
Author’s response: Thank you for this interesting question. We have provided 2 new supplementary figures assessing the data for any trends related to age and sex. Supplementary figure 3 shows no observed sub-clustering by age group for training samples, and supplementary figure 4 also shows no sex-related trends. We also added these findings to our results text (paragraph 3).